# Vitamin D Status, Vitamin D Receptor Polymorphisms, and the Risk of Incident Rosacea: Insights from Mendelian Randomization and Cohort Study in the UK Biobank

**DOI:** 10.3390/nu15173803

**Published:** 2023-08-30

**Authors:** Rui Mao, Guowei Zhou, Danrong Jing, Hong Liu, Minxue Shen, Ji Li

**Affiliations:** 1Department of Dermatology, Xiangya Hospital, Central South University, Changsha 410017, China; 218102100@csu.edu.cn (R.M.); guowei_zhou@csu.edu.cn (G.Z.); 15874884506@163.com (D.J.); 2Hunan Key Laboratory of Aging Biology, Xiangya Hospital, Central South University, Changsha 410017, China; 3National Clinical Research Center for Geriatric Disorders, Xiangya Hospital, Central South University, Changsha 410017, China; 4Hunan Engineering Research Center of Skin Health and Disease, Changsha 410008, China; 5Hunan Key Laboratory of Skin Cancer and Psoriasis, Changsha 410008, China; 6Furong Laboratory, Changsha 410008, China; 7Department of Social Medicine and Health Management, Xiangya School of Public Health, Central South University, Changsha 410017, China

**Keywords:** rosacea, 25-hydroxyvitamin D, vitamin D receptor, Mendelian randomization, UK Biobank

## Abstract

Background: Previous cross-sectional studies have failed to definitively establish a causal relationship between serum 25-hydroxyvitamin D (25OHD) concentrations and the onset of rosacea. Objective: To investigate the potential association between serum 25OHD levels, vitamin D receptor (VDR) polymorphisms, and the risk of developing incident rosacea. Methods: This cross-sectional population-based cohort study utilizing 370,209 individuals from the UK Biobank. Cox proportional hazard regression models and two-sample Mendelian randomization (MR) analyses were applied to explore the causative relationship between 25OHD and incident rosacea. Results: Our findings revealed that elevated levels of serum 25OHD were inversely correlated with the risk of incident rosacea. Specifically, compared to participants with 25OHD levels below 25 nmol/L, the multivariate-adjusted HR for incident rosacea was 0.81 (95% CI: 0.70, 0.94) in those with 25OHD levels exceeding 50 nmol/L. Further, in comparison to individuals with serum 25OHD less than 25 nmol/L and the rs731236 (*TaqI*) AA allele, those with serum 25OHD higher than 75 nmol/L and the *TaqI* GG allele had a multivariate-adjusted HR of 0.51 (95% CI 0.32 to 0.81) for developing rosacea. Results from the MR study supported a significant association, with each standard deviation increase in serum 25OHD concentrations correlating to a 23% reduced risk of rosacea (HR = 0.77, 95% CI: 0.63, 0.93). Conclusions: The findings of this cohort study indicate an inverse association between increased concentrations of serum 25OHD and the risk of developing incident rosacea. While our results highlight the potential protective role of vitamin D, the definitive efficacy of vitamin D supplementation as a preventive strategy against rosacea requires further investigation.

## 1. Introduction

Rosacea is characterized as a chronic and frequently relapsing inflammatory skin disorder. Gether et al.’s meta-analysis indicates that approximately 5.46% of the global population is affected by rosacea [1]. Due to its “drunken” appearance and sensitive symptoms such as burning and tingling, the impact of the disease on the life and psychology of patients cannot be ignored [2,3]. Therefore, the early identification of modifiable risk factors to prevent or mitigate the onset and progression of rosacea is of vital public health importance.

Serum 1,25(OH)2D3 significantly affects the innate and adaptive immune system by regulating the expression of antimicrobial peptides [4,5,6]. However, there is limited evidence regarding the relationship between vitamin D status and rosacea. Some cross-sectional studies have shown that serum vitamin D levels in rosacea patients are lower than those in healthy controls [7]. Still, there are also studies suggesting that vitamin D levels are higher in rosacea patients [8,9]. So far, no prospective studies have shown the relationship between vitamin D status and rosacea.

Vitamin D receptor (VDR) is widely found in the human body and has several polymorphic variants, including single nucleotide polymorphism (SNP) BsmI, ApaI, FokI, and TaqI [10]. The VDR functions as a ligand-activated transcription factor, orchestrating gene regulation via vitamin D response elements proximal to the initiation sites of target genes [11]. The gene GC encodes the vitamin D binding protein, a pivotal component in vitamin D transport and metabolism. This protein acts as the primary plasma transporter for both vitamin D and its respective metabolites [12]. Integral to the vitamin D pathway is an ensemble of cytochrome P450-associated sterol hydroxylases. These enzymes facilitate both the synthesis and catabolism of the active hormone, which subsequently serves as the ligand for VDR-mediated transcriptional modulation [13]. Cross-sectional studies have shown significant differences in VDR polymorphisms between rosacea patients and normal controls [9]. However, whether genetic variants in VDR alter the relationship between serum 25OHD and the risk of rosacea remains unclear.

In the current study, our primary aim was to prospectively assess the association of serum 25OHD concentrations and VDR polymorphisms with the risk of incident rosacea among participants from the UK Biobank, aiming to fill existing knowledge gaps. As secondary objectives, we further explored the interactions between serum 25OHD levels and various factors, including age, gender, BMI, degree of insomnia, vitamin D supplementation, physical activity level, and duration of sun exposure during summer, and how they might collectively influence the risk of developing rosacea.

## 2. Materials and Methods

This study utilized data from the UK Biobank (UKB) under project number 55257. All participants signed informed consent forms, and the study received ethical approval from the Northwest Multi-Center Research Ethics Committee (London, UK). Our research was conducted in alignment with the Strengthening the Reporting of Observational Studies in Epidemiology (STROBE) reporting guideline.

### 2.1. Study Design and Population

Several studies have described the study design and methodology for the UK Biobank study in detail [14,15]. Briefly, the UK Biobank recruited half a million community volunteers aged 40–69 across the UK between 2006 and 2010. Participants provided detailed health information through touch-screen questionnaires and direct anthropometry. Blood samples were collected for genotyping and biomarker analysis. In this study, we excluded patients with rosacea disease at baseline and patients with a serum 25OHD concentration missing value at baseline. For genotype analysis, we excluded non-European ancestry and related participants. A total of 370,209 participants were included in the final analysis after the further exclusion of participants with excessive heterozygosity and highly absent or mismatched sex. The whole screening flow chart of this study is shown in Appendix A.

### 2.2. Assessment of Serum 25OHD

Serum vitamin D (nanomoles per litre) was measured by chemiluminescence technology analysis on a DiaSorin Ltd., LIAISON XL (Saluggia, Italy). The measurements were calibrated and quality-controlled by the UK Biobank. Briefly, they provided coefficients of variation for samples with low, medium, and high levels of each biomarker. The coefficient of variation of serum 25OHD was 5.0~6.1%. In addition, the external quality assurance results for vitamin D were 100%. The UK Biobank website shows the measurements’ details [16].

### 2.3. Assessment of Rosacea

The diagnosis of rosacea was confirmed mainly by hospital records obtained from the England Hospital Morbidity Statistics, the Wales Patient Morbidity Database and the Scottish Morbidity Record data. Additional cases were supplemented by looking at self-reported primary care and death registration data. The diagnosis of rosacea was recorded using the International Classification of Diseases Coding System (ICD-10: L71) [17]. Patients who self-reported or identified rosacea through hospital admission records at baseline were excluded. Each person was followed up from the assessment date until 1 March 2022.

### 2.4. Polymorphisms in VDR

All 500,000 UK Biobank participants provided samples of blood for long-term storage and analysis, including genetic, when they volunteered for the project from 2006 to 2010. US-based Affymetrix undertook the original genotyping of samples in 2013–2014. The VDR polymorphisms were detected using the SNP chip from the UKB, which encompasses specific loci associated with VDR polymorphisms. Four single nucleotide polymorphisms {rs2228570 (*FokI*), rs7975232 (*ApaI*), rs731236 (*TaqI*), and rs1544410 (*BsmI*) [18]} were included in this study to explore the association between VDR and rosacea.

### 2.5. Assessment of Covariates

Information on socio-demographic status, lifestyle and dietary factors, medical history, and medication at baseline assessment was obtained using a touch-screen, self-filled questionnaire. A nurse measured height and weight, and body mass index (BMI) was calculated. Physical activity, including self-reported moderate-intensity and high-intensity activities, was measured in MET minutes per week.

### 2.6. Statistical Analysis

According to the Endocrine Society Clinical Practice Guidelines [19], serum 25OHD was categorized into 4 groups: severely deficient (<25 nmol/L), moderately deficient (25 to <50 nmol/L), insufficient (50 to <75 nmol/L), and optimal (≥75 nmol/L). Baseline characteristics were reported as mean (SD, continuous) or n (%, categorical variable) based on baseline serum 25OHD concentration intervals. We also divided the serum 25OHD levels into 4 quantiles {Q1 (<33.5 nmol/L), Q2 (33.5 to 48.0 nmol/L), Q3 (48.0 to 63.4 nmol/L), Q4 (>63.4 nmol/L)}. Each participant was followed up from the baseline date to the date of rosacea diagnosis, death, or the end of follow-up, whichever came first.

Hazard ratios (HRs) and 95%CI were assessed using Cox proportional risk models. The model was adjusted by gender (male/female), age (sequential), average annual household gross income (<£18,000, £18,000–£30,999, £31,000–£51,999, £52,000–£100,000, >£100,000, and “not known” or missing), season of blood collection (winter: December–February; spring: March–May; summer: June–August; and autumn: September–November), education (CSEs or equivalent, A levels/AS levels or equivalent, College or University degree, National Vocational Qualification (NVQ) or Higher National Diploma (HND) or Higher National Certificate (HNC) or equivalent, O levels/General Certificate of Secondary Education (GCSEs) or equivalent, professional qualifications, and none of the above), race (mixed-European, white, South Asian, black and others), alcohol consumption (daily, month to week, or never), Townsend Deprivation Index (continuous), BMI (kilograms per square meter, continuous), smoking status (current, previous, or never), physical activity (MET minutes per week, consecutive), time out summer (hours, consecutive), insomnia (never, sometimes, or usually), and vitamin D supplementation (yes or no).

We used a restricted cubic spline model with four knots to explore the dose–response relationship between serum 25OHD and incident rosacea. The analysis was also stratified by sex (male and female), age (<60 y and ≥60 y), BMI (<30 kg/m^2^ and ≥30 kg/m^2^), physical activity level (<120 and ≥120 MET minutes per week), 25OHD supplementation or not, and insomnia level (never, sometimes, and usually). We also performed several sensitivity analyses to validate our results’ robustness. First, we excluded participants who developed rosacea within 2 years of follow-up to minimize the possibility of reverse causality in the observed associations. Second, we made additional adjustments to the use of vitamin D supplements. In addition, we adjusted the sleeplessness status to test the potential mediating effect of insomnia.

All analyses were performed using R software, version 4.2.0. Two-side *p* < 0.05 was considered as the threshold with statistical significance.

## 3. Results

Among the 370,209 participants, the overall mean 25OHD concentration was 49.63 nmol/L (SD = 21.07). Baseline characteristics comparisons across different serum vitamin D ranges are presented in Table 1. Individuals with elevated 25OHD levels were primarily British, older, and exhibited a reduced Townsend deprivation index. They also presented with lower BMIs, higher likelihood of regular alcohol consumption, reduced smoking prevalence, more sun exposure during summer, frequent active vitamin D supplementation, and increased physical activity levels. The overall prevalence of rosacea in the UKB database was 1.22%. Over an average follow-up of 13.22 years, 1938 incident rosacea cases were documented, comprising 1509 (77.9%) cases in primary care and 429 (22.1%) in hospital admissions. Cox proportional risk regression analysis suggested an inverse correlation between higher serum 25OHD concentrations and the risk of incident rosacea (Table 2), both in unadjusted and multivariate-adjusted models. Compared with participants with serum 25OHD greater than 25 nmol/L, participants with serum 25OHD greater than 50 nmol/L had a multivariate-adjusted HR of 0.81 (95%CI 0.70 to 0.94, *P_trend_* = 0.01) for incident rosacea. When the serum 25OHD concentration was divided by quartile, participants in the fourth quartile had a 20% lower risk of developing rosacea than those in the first. In addition, the risk of incident rosacea was reduced by 12% for each unit increase in the natural log converted serum concentration of 25OHD (Table 2). Analysis of the restricted cubic spline model also confirmed a linear dose–response relationship between serum 25OHD concentration and the risk of incident rosacea (Appendix A, all *P_linearity_* = 0.0027).

As for genetic analysis, the serum 25OHD concentration of rs7975232 (*ApaI*) and rs2228570 (*FokI*) mutant (GG and AA) was significantly lower than that of the wild type (AA and CC) (Appendix A, *p* < 0.05). There was no significant difference in serum 25OHD concentrations between the wild, heterozygous and mutant genotypes of rs731236 (*TaqI*) and rs1544410 (*BsmI*). However, after multivariate correction, subjects with rs731236 allele AG and GG had a 10% and 12% lower risk of rosacea than the AA allele (Appendix A, *p* < 0.05). The association between serum 25OHD concentration and rs731236 gene variation and the risk of incident rosacea is shown in Figure 1 and Appendix A. In the GG allele of rs731236 (*TaqI*), participants with serum 25OHD concentration greater than 75 nmol/L had a 51% lower risk of incident rosacea than those with serum 25OHD concentration greater than 25 nmol/L (HR = 0.49, 95%CI 0.29 to 0.84, *p* = 0.010, *p_trend_* = 0.042). In addition, compared with participants with serum 25OHD less than 25 nmol/L and the allele of rs731236 (*TaqI*) AA, those with serum 25OHD greater than 75 nmol/L and the allele of rs731236 (*TaqI*) GG had a multivariate-adjusted HR of 0.51 (95%CI 0.32 to 0.81, *p* = 0.004, *p_trend_* = 0.005) for incident rosacea. Interaction analysis suggested no significant interaction between serum 25OHD concentration and VDR polymorphism (Appendix A).

We also analyzed the interaction of covariates such as age, sex, BMI, insomnia degree, 25OHD supplement, physical activity level, and summer sunshine duration with serum 25OHD (Appendix A). The results suggest a significant interaction between sex and 25OHD (*p_interaction_* = 0.032). Stratified analysis by sex, age, BMI, physical activity level, 25OHD supplementation or not, and insomnia level suggested that after multivariate adjustment the effect of serum 25OHD concentration on rosacea onset was more significant in participants who were younger than 60 years old (HR = 0.74, 95%CI 0.57 to 0.95) or male (HR = 0.68, 95%CI 0.51 to 0.92) (Appendix A). Cumulative risk curve model analysis also confirms our results (Appendix A). Further stratified analysis suggested that among male participants younger than 60 years old, the multivariate corrective HR for incident rosacea was 0.63 (*p_trend_* = 0.012) for serum 25OHD concentrations greater than 50 nmol/L (95%CI 0.46 to 0.86) and 75 nmol/L (95%CI 0.41 to 0.98), compared with those greater than 25 nmol/L. In addition, in male participants younger than 60 years old, the risk of incident rosacea was reduced by 21% for each unit increase in natural logarithmic converted serum 25OHD concentration (Appendix A). In addition, stratified analysis by sex suggested that VDR polymorphism had no sex specificity on rosacea onset after multivariate adjustment (Appendix A). No cumulative effect of serum 25OHD on incident rosacea was observed in male participants with GG allele of rs731236 (*TaqI*) (Appendix A).

The results of sensitive analyses suggested that adjusted HRs and 95% CI for serum 25OHD levels among incident rosacea patients after excluding outcomes that occurred within two years of follow-up (HR = 0.76, 95%CI 0.61 to 0.93) were more significant than before the exclusion (Appendix A). In addition, the effect of serum 25OHD on the onset of rosacea was more significant after further exclusion of participants who had used vitamin D (HR = 0.71, 95%CI 0.56 to 0.90) and then those who had frequent sleeplessness (HR = 0.67, 95%CI 0.51 to 0.90).

We further performed a two-sample MR analysis to validate the causal relationship between serum 25OHD concentration and incident rosacea. The detailed method of MR analysis and the results of sensitivity analysis are provided in eMethod (Appendix A) [20,21,22,23,24]. The results of GCST010144 confirmed that a higher serum 25OHD concentration was a protective factor for rosacea pathogenesis (*p* < 0.05). Although the *p*-value of ebi-a-GCST90000615 was greater than 0.05, the meta-analysis after MR suggested that 25OHD had a protective causal relationship to rosacea among the five methods. The forest plot is presented on Figure 2.

## 4. Discussion

In this large prospective study of incident rosacea patients, we found that a higher serum 25OHD concentration was significantly associated with a lower risk of incident rosacea. A significant linear dose–response relationship was also observed between serum 25OHD (range 10 to 340 nmol/L) and rosacea. Participants with different VDR polymorphisms had a significantly different risk of incident rosacea. A series of stratified and sensitivity analyses demonstrated the robustness of these results. MR analysis confirmed a causal relationship between serum 25OHD and the risk of incident rosacea.

Evidence on the relationship between vitamin D status and rosacea development is limited and inconsistent in patients with rosacea with vitamin D deficiency or insufficiency. It is important to note that all of these studies were cross-sectional designs, limited by small sample sizes, and not accounting for confounding factors such as physical activity, sun exposure, and sleeplessness [7,8,9]. Based on a large sample size and a long follow-up period, our study is the first prospective study to demonstrate that a higher serum 25OHD concentration is significantly associated with a lower risk of incident rosacea. The overall prevalence of rosacea in the UKB database was 1.22%, congruent with previous UK studies [25]. The results were independent of significant confounding factors, including time spent outdoors in summer, sleep status, physical activity, BMI, and vitamin supplement status. In addition, we demonstrated a linear dose–response relationship between serum 25OHD and incident rosacea. The further stratified analysis found that the protective effect of a high serum 25OHD concentration against incident rosacea was more significant in male or young participants, especially young male participants. Last but not least, we demonstrated a negative causal relationship between serum 25OHD levels and incident rosacea onset using MR analysis for the first time.

Similar to the conclusion of previous studies [26], some wild-type homozygotes of VDR (*Apal* and *Fokl*) had higher serum 25OHD concentrations than heterozygotes and mutant homozygotes. In addition, there is limited evidence on VDR polymorphism and rosacea pathogenesis. Only one cross-sectional study involving 120 patients has been reported [9]. The researchers found that heterozygous and mutated *ApaI* polymorphisms were associated with an increased risk of rosacea, while mutated *TaqI* polymorphisms were associated with a reduced risk of rosacea. It is worth noting that the cross-sectional study design cannot definitively establish cause-and-effect relationships. Limited sample size and unknown confounding variables may also bias the results. In the current study, participants who carried the GG allele of rs731236 (*TaqI*) were less likely to develop rosacea. The protective effect of a high serum 25OHD concentration against rosacea onset was even more significant in participants with the GG allele of rs731236 (*TaqI*). In addition, subjects with serum 25OHD greater than 75 nmol/L and the allele of rs731236 (*TaqI*) GG had a significantly reduced risk of incident rosacea compared to participants with serum 25OHD less than 25 nmol/L and the allele of rs731236 (*TaqI*) AA, and no significant interactions were observed. More prospective studies are needed to confirm these findings.

Some potential biological mechanisms may underlie the effect of serum 25OHD on the pathogenesis of rosacea. First, vitamin D3 regulates adaptive skin immunity. VDR was found to be expressed in most cells of the adaptive immune system, and the addition of 25OHD inhibited T lymphocyte proliferation and cytokine secretion in vitro [27,28]. Second, vitamin D3 regulates the skin’s innate defense barrier. Activated 25OHD promotes the expression of various antimicrobial peptides, including catheterin, which promotes the formation of a chemical protective layer on the skin surface [29,30,31]. Third, 25OHD can inhibit angiogenesis [32] and vascular dilatation caused by oxidative stress and nitric oxide [33,34]. Although some studies have shown that activated ductin fragments can easily induce rosacea [35], the relationship between the pathogenesis of rosacea and 25OHD needs to be further explored.

## 5. Strengths and Limitations

Our study has several advantages. First, it is the first prospective study to assess serum 25OHD on the risk of incident rosacea, which contains a large sample size and well-validated measures of serum 25OHD. Second, we simultaneously evaluated the influence of serum 25OHD concentration and VDR polymorphism on incident rosacea, investigated their intrinsic relationship, and stratified the most affected population. Third, we carefully adjusted for potential confounding factors such as sex, age, BMI, summer outdoor time, sleep status, active vitamin D supplementation, and exercise levels. Finally, we used MR analysis for the first time to investigate and confirm the causal relationship between serum 25OHD and rosacea.

But we should also take into account several limitations. First, based on this observational study design, there is no way to establish the most robust cause-and-effect relationships that rigorous randomized controlled trials do. Second, measurements of serum 25OHD from baseline may not be representative of long-term levels. However, some studies suggest that baseline measurements of 25OHD alone can be a reliable proxy for vitamin D status [36]. Third, most of the participants in our study were white British, who tended to have higher serum levels of 25OHD, and vitamin D metabolism varied by race and ethnicity [37]. That may limit the generality of the findings. Fourth, the current study included only participants aged 40 to 69. Since rosacea is more common in young adults [38], a selection bias may exist. The relationship between serum 25OHD and VDR polymorphism in other age groups and the incidence of rosacea needs further study. In addition, the confirmed rosacea patients did not have subtypes to further analyze the relationship between serum 25OHD and various clinical subtypes of rosacea. Fifth, given that 85% to 90% of circulating vitamin D binds to vitamin D-binding proteins and that bioactive vitamin D only accounts for a tiny fraction [39], more research is needed to evaluate the relationship between different vitamin D metabolites, including the bioavailability of 25OHD and vitamin D-binding proteins and the risk of incident rosacea in multi-ethnic populations. Sixth, there may be a possibility of a missing diagnosis of rosacea among participants in UKB, and the reliability of diagnosis is not very stable. Finally, although we have adjusted for potential confounders to the extent possible, residual confounders or unmeasured factors cannot be excluded entirely.

## 6. Conclusions

In conclusion, our study has illuminated a significant association between a higher serum 25OHD concentration and a decreased risk of incident rosacea. This relationship appears to be modulated by factors such as gender, age, and the presence of specific VDR polymorphisms. However, the causal nature of this association, including whether vitamin D supplementation might be an effective preventative measure against rosacea, remains unclear. Future large and well-designed randomized controlled trials are essential to explore the potential therapeutic implications of these findings.

## Figures and Tables

**Figure 1 nutrients-15-03803-f001:**
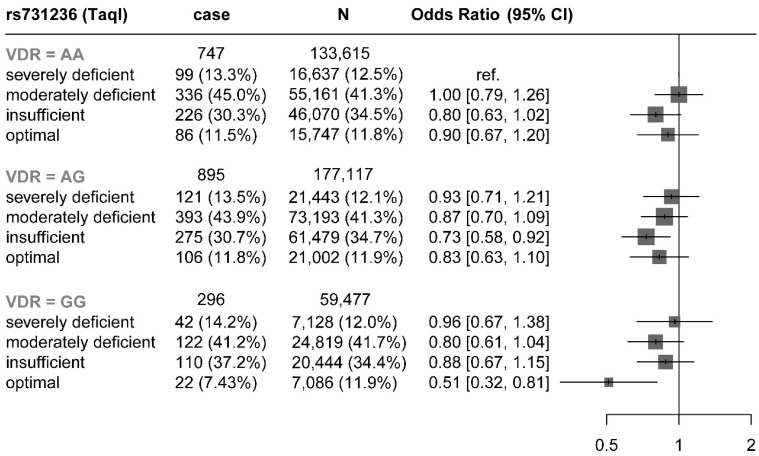
The joint associations of serum 25OHD and genetic variants in VDR with risk of rosacea. HR was adjusted by gender (male/female), age (sequential), average annual household gross income (<£18,000, £18,000–£30,999, £31,000–£51,999, £52,000–£100,000, >£100,000, and “not known” or missing), the season of blood collection, education (CSEs or equivalent, A levels/AS levels or equivalent, College or University degree, NVQ or HND or HNC or equivalent, O levels/GCSEs or equivalent, professional qualifications, and none of the above), race (mixed-European, white, South Asian, black and others), alcohol consumption (daily, month to week, or never), Townsend Deprivation Index (continuous), BMI (kilograms per square meter, continuous), smoking status (current, previous, or never), physical activity (MET minutes per week, consecutive), time out summer (hours, consecutive), insomnia (never, sometimes, or usually), and vitamin D supplementation (yes or no). ref, reference.

**Figure 2 nutrients-15-03803-f002:**
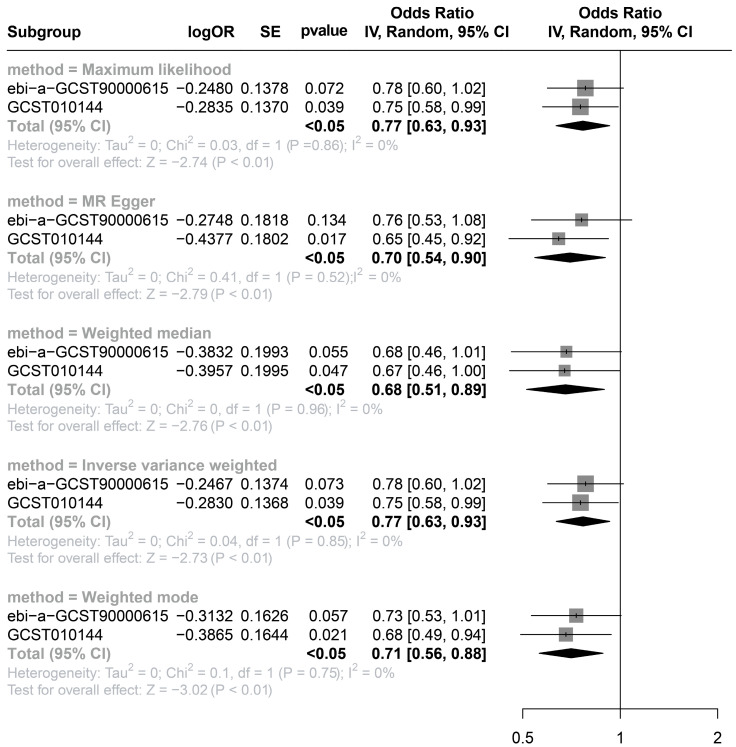
The forest plot for two-sample Mendelian randomization and meta-analysis of serum 25OHD. SE, standard error; OR, odds ratio; CI, confidence interval.

**Table 1 nutrients-15-03803-t001:** Baseline characteristics according to serum 25OHD concentrations among participants in the UK Biobank.

Variable	Overall	Severely Deficient	Moderately Deficient	Insufficient	Optimal	*p*-Value ^1^
	*N = 370,209*	*N = 45,208*	*N = 153,173*	*N = 127,993*	*N = 43,835*	
Townsend, Mean(SD)	−1.54 (2.94)	−0.63 (3.33)	−1.42 (2.98)	−1.86 (2.74)	−1.93 (2.70)	<0.001
BMI (kg/m^2^), Mean(SD)	27.37 (4.74)	28.55 (5.74)	27.85 (4.89)	26.86 (4.23)	25.95 (3.83)	<0.001
Times (y), Mean(SD)	13.22 (2.01)	13.22 (2.29)	13.25(2.02)	13.21 (1.92)	13.16 (1.95)	<0.001
status, n (%)						0.007
Incident rosacea	1938 (0.5%)	262 (0.6%)	851 (0.6%)	611 (0.5%)	214 (0.5%)	
Rosacea-free	368,271 (99%)	44,946 (99%)	152,322 (99%)	127,382 (100%)	43,621 (100%)	
education, n (%)						<0.001
CSEs or equivalent	20,720 (5.6%)	2459 (5.4%)	8333 (5.4%)	7143 (5.6%)	2785 (6.4%)	
A levels/AS levels or equivalent	42,200 (11%)	5257 (12%)	17,500 (11%)	14,621 (11%)	4822 (11%)	
College or University degree	118,709 (32%)	15,908 (35%)	51,700 (34%)	38,982 (30%)	12,119 (28%)	
none of the above	62,287 (17%)	7548 (17%)	24,785 (16%)	22,014 (17%)	7940 (18%)	
NVQ or HND or HNC or equivalent	25,104 (6.8%)	2947 (6.5%)	10,191 (6.7%)	8868 (6.9%)	3098 (7.1%)	
O levels/GCSEs or equivalent	82,196 (22%)	8984 (20%)	32,895 (21%)	29,519 (23%)	10,798 (25%)	
professional qualifications	18,993 (5.1%)	2105 (4.7%)	7769 (5.1%)	6846 (5.3%)	2273 (5.2%)	
sex, n (%)						0.5
female	195,341 (53%)	23,850 (53%)	80,605 (53%)	67,731 (53%)	23,155 (53%)	
male	174,868 (47%)	21,358 (47%)	72,568 (47%)	60,262 (47%)	20,680 (47%)	
Time out summer, Mean(SD)	3.81 (2.39)	3.33 (2.32)	3.65 (2.35)	4.00 (2.38)	4.30 (2.43)	<0.001
Ethnic background, n (%)						<0.001
British	362,177 (98%)	41,763 (92%)	149,733 (98%)	127,018 (99%)	43,663 (100%)	
Caribbean	3248 (0.9%)	1178 (2.6%)	1561 (1.0%)	436 (0.3%)	73 (0.2%)	
Indian	3945 (1.1%)	2078 (4.6%)	1505 (1.0%)	320 (0.3%)	42 (<0.1%)	
White	363 (<0.1%)	58 (0.1%)	161 (0.1%)	107 (<0.1%)	37 (<0.1%)	
White and Black Caribbean	476 (0.1%)	131 (0.3%)	213 (0.1%)	112 (<0.1%)	20 (<0.1%)	
sleeplessness, n (%)						<0.001
never	90,406 (24%)	10,922 (24%)	37,730 (25%)	31,145 (24%)	10,609 (24%)	
sometimes	176,065 (48%)	21,204 (47%)	72,821 (48%)	61,325 (48%)	20,715 (47%)	
usually	103,738 (28%)	13,082 (29%)	42,622 (28%)	35,523 (28%)	12,511 (29%)	
Smoking status, n (%)						<0.001
current	37,463 (10%)	7509 (17%)	15,869 (10%)	10,388 (8.1%)	3697 (8.4%)	
never	203,000 (55%)	23,633 (52%)	84,589 (55%)	71,191 (56%)	23,587 (54%)	
previous	129,746 (35%)	14,066 (31%)	52,715 (34%)	46,414 (36%)	16,551 (38%)	
Alcohol intake frequency, n (%)						<0.001
daily	78,418 (21%)	8808 (19%)	30,886 (20%)	28,007 (22%)	10,717 (24%)	
month to week	227,989 (62%)	25,171 (56%)	94,248 (62%)	81,032 (63%)	27,538 (63%)	
never	63,802 (17%)	11,229 (25%)	28,039 (18%)	18,954 (15%)	5580 (13%)	
Age at recruitment, Mean(SD)	56.65 (8.07)	55.09 (8.09)	56.34 (8.07)	57.34 (7.97)	57.30 (8.04)	<0.001
income, n (%)						<0.001
18,000 to 30,999	95,780 (26%)	10,909 (24%)	39,038 (25%)	34,004 (27%)	11,829 (27%)	
31,000 to 51,999	97,560 (26%)	11,547 (26%)	40,685 (27%)	33,784 (26%)	11,544 (26%)	
52,000 to 100,000	75,668 (20%)	8905 (20%)	31,970 (21%)	25,948 (20%)	8845 (20%)	
greater than 100,000	19,890 (5.4%)	2111 (4.7%)	8248 (5.4%)	7042 (5.5%)	2489 (5.7%)	
less than 18,000	81,311 (22%)	11,736 (26%)	33,232 (22%)	27,215 (21%)	9128 (21%)	
MET, Mean(SD)	44.77 (45.34)	37.59 (42.08)	42.24 (43.96)	47.73 (46.28)	52.38 (48.69)	<0.001
25OHD supplement, n (%)	67,728 (18%)	3890 (8.6%)	23,392 (15%)	29,149 (23%)	11,297 (26%)	<0.001
Serum 25OHD (nmol/L), Mean(SD)	49.63 (21.07)	19.46 (3.85)	38.01 (7.09)	61.11 (6.95)	87.82 (12.79)	<0.001
25OHD quantile (nmol/L), n (%)						<0.001
Q1 (<33.5)	92,710 (25%)	45,208 (100%)	47,502 (31%)	0 (0%)	0 (0%)	
Q2 (33.5 to 48.0)	92,818 (25%)	0 (0%)	92,818 (61%)	0 (0%)	0 (0%)	
Q3 (48.0 to 63.4)	92,607 (25%)	0 (0%)	12,853 (8.4%)	79,754 (62%)	0 (0%)	
Q4 (>63.4)	92,074 (25%)	0 (0%)	0 (0%)	48,239 (38%)	43,835 (100%)	
taql, n (%)						0.018
GG	59,477 (16%)	7128 (16%)	24,819 (16%)	20,444 (16%)	7086 (16%)	
AG	177,117 (48%)	21,443 (47%)	73,193 (48%)	61,479 (48%)	21,002 (48%)	
AA	133,615 (36%)	16,637 (37%)	55,161 (36%)	46,070 (36%)	15,747 (36%)	
apal, n (%)						0.3
AA	103,294 (28%)	12,762 (28%)	42,870 (28%)	35,426 (28%)	12,236 (28%)	
AC	184,352 (50%)	22,409 (50%)	76,280 (50%)	63,847 (50%)	21,816 (50%)	
CC	82,563 (22%)	10,037 (22%)	34,023 (22%)	28,720 (22%)	9783 (22%)	
bsml, n (%)						0.4
TT	61,293 (17%)	7497 (17%)	25,574 (17%)	20,974 (16%)	7248 (17%)	
TC	178,118 (48%)	21,657 (48%)	73,607 (48%)	61,740 (48%)	21,114 (48%)	
CC	130,798 (35%)	16,054 (36%)	53,992 (35%)	45,279 (35%)	15,473 (35%)	
fokl, n (%)						<0.001
GG	141,007 (38%)	17,894 (40%)	58,231 (38%)	48,326 (38%)	16,556 (38%)	
AG	174,736 (47%)	20,948 (46%)	72,536 (47%)	60,683 (47%)	20,569 (47%)	
AA	54,466 (15%)	6366 (14%)	22,406 (15%)	18,984 (15%)	6710 (15%)	

^1^ Kruskal–Wallis rank sum test; Pearson’s chi-squared test.

**Table 2 nutrients-15-03803-t002:** Adjusted HRs and 95% CI for serum 25OHD levels with rosacea in the UK Biobank study.

Variables	Person-Years	Incident Rate	Incident Rosacea	Unadjusted Model		Fully Adjusted Models	
				HR (95%CI)	*p*-Value	HR (95%CI)	*p*-Value
Serum 25OHD (nmol/L)							
severely deficient	597,727	0.438	262 (13.5%)	Ref.	Ref.	Ref.	Ref.
moderately deficient	2,030,024	0.419	851 (43.9%)	0.95 [0.83;1.09]	0.495	0.94 [0.82;1.08]	0.383
insufficient	1,690,778	0.361	611 (31.5%)	0.82 [0.71;0.95]	0.007	0.81 [0.70;0.94]	0.006
optimal	577,049	0.371	214 (11.0%)	0.84 [0.70;1.00]	0.056	0.83 [0.69;1.01]	0.059
				p.trend	0.006	p.trend	0.010
25OHD (nmol/L) quantile							
Q1 (<33.5)	1,228,324	0.444	545 (28.1%)	Ref.	Ref.	Ref.	Ref.
Q2 (33.5 to 48.0)	1,229,421	0.404	497 (25.6%)	0.91 [0.80;1.03]	0.120	0.90 [0.79;1.02]	0.090
Q3 (48.0 to 63.4)	1,223,889	0.376	460 (23.7%)	0.84 [0.74;0.95]	0.007	0.84 [0.74;0.95]	0.006
Q4 (>63.4)	1,213,944	0.359	436 (22.5%)	0.80 [0.71;0.91]	0.001	0.80 [0.70;0.91]	<0.001
				p.trend	0.003	p.trend	0.006
25OHD (per SD)	4,895,578	0.396	1938	0.89 [0.83;0.95]	<0.001	0.88 [0.82;0.95]	<0.001

The model was adjusted by gender (male/female), age (sequential), average annual household gross income (<£18,000, £18,000–£30,999, £31,000–£51,999, £52,000–£100,000, >£100,000, and “not known” or missing), the season of blood collection, education (CSEs or equivalent, A levels/AS levels or equivalent, College or University degree, NVQ or HND or HNC or equivalent, O levels/GCSEs or equivalent, professional qualifications, and none of the above), race (mixed-European, white, South Asian, black and others), alcohol consumption (daily, month to week, or never), Townsend Deprivation Index (continuous), BMI (kilograms per square meter, continuous), smoking status (current, previous, or never), physical activity (MET minutes per week, consecutive), time out summer (hours, consecutive), insomnia (never, sometimes, or usually), and vitamin D supplementation (yes or no). HR, hazard ratio; CI, confidence interval; 25OHD, 25-hydroxyvitamin D.

## Data Availability

This work was conducted using the UK Biobank Resource (Application Number: 55257, with extended scope). The UK Biobank is an open-access resource, and bona fide researchers can apply to use the UK Biobank dataset by registering and applying at http://ukbiobank.ac.uk/register-apply/ (accessed on 1 May 2022). Further information is available from the corresponding author upon request.

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
