# Peer review of "Vitamin D Status, Vitamin D Receptor Polymorphisms, and the Risk of Incident Rosacea: Insights from Mendelian Randomization and Cohort Study in the UK Biobank"

_nutrients, 2023, doi:10.3390/nu15173803_

Round 1
Reviewer 1 Report
The study “Vitamin D status, vitamin D receptor polymorphisms, and the risk of incident rosacea: insights form Mendelian randomization and Cohort study in the UK biobank” reports a significant linear dose response relationship between serum 25OHD and rosacea.
The results plainly demonstrated that participants with different VDR polymorphisms had a significantly different risk of incident rosacea. Mendelian randomization analysis confirms the causal relationship serum 25OHD and risk of incident rosacea. In a prospective study with large sample size and a long follow-up.
Additionally, this study suggest that this relationship appears to be modulated by variables such as gender, age, and specific vitamin D receptor polymorphisms.
However, some important points must be reviewed:
-Page 2/ line 53 and 54: supplementation does not math with the final conclusion (vitamins D supplementation might be an effective preventative measure against rosacea, remains unclear);
-Page 3/ line 135: item 2.4 Polymorphism on VDR might explain the methodology;
-Page 5/ line 198 -200: review this paragraph;
-Page 10/ table 2: insert HR in the subtitle;
Reviewer 2 Report
I had to revise the article entitled ” Vitamin D Status, Vitamin D Receptor Polymorphisms, and the Risk of Incident Rosacea: Insights from Mendelian Randomization and Cohort Study in the UK Biobank” submitted for publishing in Nutrients journal.
The role of vitamin D in development of acute or chronic diseases is an interesting topic that received a large attention in the last decade. The present study analyzed the relationship between serum 25OHD, vitamin D receptor polymorphisms and the incidence of Rosacea. It is an interesting research and topic, a comprehensive study with many factors analyzed which provide a clearer answer regarding this issue.
I suggest some observations:
In the introduction section the authors should introduce few aspects regarding the influence of Vitamin D Receptor Polymorphisms on serum level of vitamin D. They also should include the secondary aims, taking in consideration that they analyzed the incidence of Rosacea in relationship with other factors.
Please provide the title of the project number 55257.
Detail the method for Polymorphisms in VDR.
Comparisons with other results should be included in discussion section not at the results (see lines 209-210)
